# A New High-Throughput Tool to Screen Mosquito-Borne Viruses in Zika Virus Endemic/Epidemic Areas

**DOI:** 10.3390/v11100904

**Published:** 2019-09-27

**Authors:** Sara Moutailler, Lena Yousfi, Laurence Mousson, Elodie Devillers, Marie Vazeille, Anubis Vega-Rúa, Yvon Perrin, Frédéric Jourdain, Fabrice Chandre, Arnaud Cannet, Sandrine Chantilly, Johana Restrepo, Amandine Guidez, Isabelle Dusfour, Filipe Vieira Santos de Abreu, Taissa Pereira dos Santos, Davy Jiolle, Tessa M. Visser, Constantianus J. M. Koenraadt, Merril Wongsokarijo, Mawlouth Diallo, Diawo Diallo, Alioune Gaye, Sébastien Boyer, Veasna Duong, Géraldine Piorkowski, Christophe Paupy, Ricardo Lourenco de Oliveira, Xavier de Lamballerie, Anna-Bella Failloux

**Affiliations:** 1UMR BIPAR, Animal Health Laboratory, ANSES, INRA, Ecole Nationale Vétérinaire d’Alfort, Université Paris-Est, 94700 Maisons-Alfort, France; lenayousfi@hotmail.fr (L.Y.); elodie.devillers@anses.fr (E.D.); 2Arboviruses and Insect Vectors, Departement of Virology, Institut Pasteur, 75015 Paris, France; laurence.mousson@pasteur.fr (L.M.); marie.vazeille@pasteur.fr (M.V.); 3Laboratory of Vector Control Research, Institut Pasteur of Guadeloupe–Lieu-dit Morne Jolivière, 97139 Les Abymes, Guadeloupe, France; anubis.vega-rua@pasteur.fr; 4CNEV, IRD, 34000 Montpellier, France; yperrin@eid-med.org (Y.P.); frederic.jourdain@ird.fr (F.J.); fabrice.chandre@ird.fr (F.C.); ar.cannet@gmail.com (A.C.); 5UMR MIVEGEC, IRD-CNRS-Montpellier University, 34000 Montpellier, France; tayssadnz@gmail.com (T.P.d.S.); davy_jiolle@hotmail.com (D.J.); christophe.paupy@ird.fr (C.P.); 6Collectivité Territoriale de Guyane (CTG), Direction de la Démoustication, 97300 Cayenne, France; sandrine.chantilly@ctguyane.fr (S.C.); johana.restrepo@ctguyane.fr (J.R.); 7Unité adaptation et contrôle des vecteurs, Institut Pasteur de la Guyane, 97300 Cayenne, France; aguidez@pasteur-cayenne.fr (A.G.); isabelle.dusfour@pasteur.fr (I.D.); 8Laboratório de Mosquitos Transmissores de Hematozoários, Instituto Oswaldo Cruz, Fiocruz, Rio de Janeiro 21040900, Brazil; gigante_bio@yahoo.com.br (F.V.S.d.A.); lourenco@ioc.fiocruz.br (R.L.d.O.); 9Laboratory of Entomology, Wageningen University & Research, 6708 PB Wageningen, The Netherlands; tessa.visser@wur.nl (T.M.V.); sander.koenraadt@wur.nl (C.J.M.K.); 10Central Laboratory of the Bureau of Public Health, P.O. Box l863 Paramaribo, Suriname; mer_wongso@hotmail.com; 11Unité d’Entomologie Médicale, Institut Pasteur de Dakar, BP 220 Dakar, Senegal; Mawlouth.DIALLO@pasteur.sn (M.D.); diawo.diallo@pasteur.sn (D.D.); alioune.gaye@pasteur.sn (A.G.); 12Medical Entomology Platform and Virology Unit, Institut Pasteur du Cambodge, P.O Box. 983 Phnom Penh, Cambodia; sboyer@pasteur-kh.org (S.B.); dveasna@pasteur-kh.org (V.D.); 13Unité des Virus Emergents (UVE), Aix Marseille Université, IRD 190, INSERM 1207, IHUMéditerranée Infection, 13385 Marseille, France; geraldine.piorkowski@inserm.fr (G.P.); xavier.de-lamballerie@univ-amu.fr (X.d.L.)

**Keywords:** mosquito-borne viruses, molecular epidemiology, surveillance, microfluidic analysis

## Abstract

Mosquitoes are vectors of arboviruses affecting animal and human health. Arboviruses circulate primarily within an enzootic cycle and recurrent spillovers contribute to the emergence of human-adapted viruses able to initiate an urban cycle involving anthropophilic mosquitoes. The increasing volume of travel and trade offers multiple opportunities for arbovirus introduction in new regions. This scenario has been exemplified recently with the Zika pandemic. To incriminate a mosquito as vector of a pathogen, several criteria are required such as the detection of natural infections in mosquitoes. In this study, we used a high-throughput chip based on the BioMark™ Dynamic arrays system capable of detecting 64 arboviruses in a single experiment. A total of 17,958 mosquitoes collected in Zika-endemic/epidemic countries (Brazil, French Guiana, Guadeloupe, Suriname, Senegal, and Cambodia) were analyzed. Here we show that this new tool can detect endemic and epidemic viruses in different mosquito species in an epidemic context. Thus, this fast and low-cost method can be suggested as a novel epidemiological surveillance tool to identify circulating arboviruses.

## 1. Introduction

The World Health Organization stated in February 2016 that Zika infection was considered as a public health emergency of international concern [1] opening a new chapter in the history of vector-borne diseases. Arboviruses are viruses transmitted among vertebrate hosts by arthropod vectors. Successful transmission of an arbovirus relies on a complex life cycle in the vector, which after midgut infection and dissemination, is released in saliva for active transmission to the vertebrate host [2]. Arboviruses belong to nine families: Asfarviridae, Flaviviridae, Orthomyxoviridae, Reoviridae, Rhabdoviridae, the newly recognized Nyamiviridae (order Mononegavirales), and the families Nairoviridae, Phenuiviridae, and Peribunyaviridae in the new order, Bunyavirales. Most arboviruses possess an RNA genome and are mainly transmitted by mosquitoes [3]. While acute infections in vertebrate hosts are typically self-limiting, arboviruses establish persistent infections in arthropods granting a central role as a viral reservoir to the vector [4].

Arboviruses circulate primarily within an enzootic cycle involving zoophilic vector species and non-human hosts. Recurrent spillovers cause occasional infections of humans initiating an epidemic cycle. Arboviruses such as dengue (DENV; *Flavivirus*, Flaviviridae), chikungunya (CHIKV; *Alphavirus*, Togaviridae), Zika (ZIKV; *Flavivirus*, Flaviviridae), and Yellow fever virus (YFV; *Flavivirus*, Flaviviridae) do not need to amplify in wild animals to cause outbreaks in humans, which act simultaneously as amplifier, disseminator, and source of infection for the major vectors, the anthropophilic mosquitoes *Aedes aegypti* and *Aedes albopictus* [5]. Thus, the success of these viruses comes from their feature to be mainly transmitted by human-biting mosquitoes strongly adapted to urban environments. The establishment of a new epidemic cycle is undoubtedly related to the introduction of a viremic vertebrate host (humans, animals) acting as a vehicle for importation of the virus into environments receptive to viral amplification. Other arboviruses such as West Nile virus (WNV; *Flavivirus*, Flaviviridae) remain circulating within an enzootic cycle with sporadic spillovers causing human cases.

Many regions experience simultaneous circulation of different arboviruses [6,7], and co-infections in vectors were reported [8]. These coinfections can present an opportunity for viruses to exchange genetic material. Impacts of such genetic events on virulence for vertebrate hosts are still unknown [9]. Thus, being able to detect a wide range of arboviruses in thousands of field-collected mosquitoes in a single experiment can be a valuable tool to predict arboviral emergences in human populations. Indeed, similar methods were developed with success to screen tick-borne pathogens (bacteria, parasites, and viruses) and allowed the detection of expected and unexpected pathogens in large-scale epidemiological studies [10,11]. Therefore, we developed a high-throughput system based on real-time microfluidic PCR, which is able to detect 96 mosquito-borne viruses in 96 samples within one single run. With this method, we have screened: (1) Mosquitoes infected artificially using a feeding system to validate our tool, (2) mosquitoes collected in countries endemic for the major human arboviruses (e.g., Senegal, Cambodia, Brazil), and (3) mosquitoes collected during the Zika and Yellow fever outbreaks in the Americas (French Guiana, Guadeloupe, Brazil, Suriname). This method allowed for the detection of epidemic viruses (ZIKV, CHIKV, YFV) but also unexpected viruses (e.g., Trivittatus virus, TVTV, *Orthobunyavirus*, Bunyaviridae) underlining the need of such a tool for early detection of emerging mosquito-borne viruses.

## 2. Materials and Methods

### 2.1. Mosquitoes

To test the ability of our assays to detect viruses present in pools of mosquitoes, 47 batches of three infected mosquitoes of the species, *Ae. aegypti*, *Ae. albopictus*, and *Cx. pipiens* (infection performed by artificial feeding system), were provided by the Institut Pasteur (Paris). Six different viruses, single or double infections, were tested in a pilot study. Briefly, batches of 60 7–10-day-old females were challenged with an infectious blood meal containing 1.4 mL of washed rabbit erythrocytes, 700 μL of viral suspension, and 1 mM of adenosine 5’-triphosphate (ATP) as a phagostimulant [12]. The blood meal was provided to mosquitoes at a titer of 10^7^ focus-forming unit (FFU)/mL using a Hemotek membrane feeding system (Hemotek Ltd., Blackburn, UK). After 20 min, fully engorged females were transferred in cardboard containers and maintained with 10% sucrose until examination.

In ZIKV-endemic and -epidemic regions from South America, Africa, and Asia (Brazil, French Guiana, Guadeloupe, Suriname, Senegal, Cambodia), adult mosquitoes were collected, identified using morphological characters, and dissected to separate abdomen from the remaining body parts (RBP) (See Tables 1–6 for details). Abdomens of the same species were grouped by pools of 20–30 individuals in cryovials, and RBP were stored individually at −80 °C until further analysis.

### 2.2. RNA Extraction

Total RNAs were extracted from each pool using the Nucleospin RNA II extraction kit (Macherey-Nagel, Hoerdt, France). Pools were ground in 350 µL Lysis Buffer and 3.5 µL β-mercaptoethanol using the homogenizer Precellys^®^24 Dual (Bertin, France) at 5500 rpm for 20 s. Total RNA per pool was eluted in 50 µL of RNase free water and stored at −80 °C until use.

When pools of abdomens were positive for virus, the RBP (head/thorax) of individual mosquitoes composing each pool were homogenized in 300 µL of DMEM with 10% fetal calf serum using the homogenizer Precellys^®^24 Dual (Bertin, France) at 5500 rpm for 20 s. Then, total RNAs were extracted from 100 µL of homogenates using the Nucleospin RNA II extract kit (Macherey-Nagel, Germany) and 200 µL were conserved at −80 °C for attempts to isolate the virus. Total RNA per sample was eluted in 50 µL of RNase free water and stored at −80 °C until use.

### 2.3. Reverse Transcription and cDNA Pre-Amplification

RNAs were transcribed to cDNA by reverse transcription using the qScript cDNA Supermix kit according to the manufacturer’s instructions (Quanta Biosciences, Beverly, USA). Briefly, the reaction was performed in a final volume of 5 µL containing 1 µL of qScript cDNA supermix 5X, 1 µL of RNA, and 3 µL of RNase free water; with one cycle at 25 °C for 5 min, one cycle at 42 °C for 30 min, and one final cycle at 85 °C for 5 min.

For cDNA pre-amplification, the Perfecta Preamp Supermix (Quanta Biosciences, Beverly, USA) was used according to the manufacturer’s instructions. All primers were pooled to a final concentration of 200 nM each. The reaction was performed in a final volume of 5 μL containing 1 μL Perfecta Preamp 5×, 1.25 μL pooled primers, 1.5 µL distilled water, and 1.25 μL cDNA, with one cycle at 95 °C for 2 min, 14 cycles at 95 °C for 10 s, and 3 min at 60 °C. At the end of the cycling program, the reactions were 1:5 diluted. Pre-amplified cDNAs were stored at −20 °C until use.

### 2.4. Assay Design

Mosquito-borne viruses (MBV), their targeted genes, and the corresponding primers/probe sets are listed in Appendix A. For a total of 64 viruses including 149 genotypes/serotypes, primers and probes were specifically designed. Indeed, selection was based on specific constraints of temperature of annealing (60 °C for primers and 70 °C for probes); primers/probe sets published in the literature were included if they fit into these criteria. Each primer/probe set was validated using a dilution range of several cDNA positive controls (when available) (Appendix A), by real-time PCR on a LightCycler^®^ 480 (LC480) (Roche Applied Science, Penzberg, Germany). Real-time PCR assays were performed in a final volume of 12 µL using the LightCycler^®^ 480 Probe Master Mix 1X (Roche Applied Science, Germany) with primers and probes at 200 nM and 2 µL of control cDNA (virus reference material) or DNA (Plasmid). Thermal cycling conditions were as follows: 95 °C for 5 min, 45 cycles at 95 °C for 10 s, and 60 °C for 15 s, and one final cooling cycle at 40 °C for 10 s.

### 2.5. High-Throughput Real-Time PCR

The BioMark™ real-time PCR system (Fluidigm, South San Francisco, CA, USA) was used for high-throughput microfluidic real-time PCR amplification using the 96.96 dynamic arrays (Fluidigm, South San Francisco, CA, USA). These chips dispensed 96 PCR mixes and 96 samples into individual wells, after which on-chip microfluidics assembled PCR reactions in individual chambers prior to thermal cycling resulting in 9216 individual reactions. Real-time PCRs were performed using FAM- and black hole quencher (BHQ1)-labeled TaqMan probes with TaqMan Gene Expression Master Mix in accordance with manufacturer’s instructions (Applied Biosystems, Foster City, CA, USA). Thermal cycling conditions were as follows: 2 min at 50 °C, 10 min at 95 °C, followed by 40 cycles of 2-step amplification of 15 sec at 95 °C, and 1 min at 60 °C. Data were acquired on the BioMark™ real-time PCR system and analyzed using the Fluidigm real-time PCR Analysis software to obtain C_t_ values (see Michelet et al. 2014 for more details [10]). Primers and probes were evaluated for their specificity against cDNA reference samples. One negative water control was included per chip. To determine if factors present in the sample could inhibit the PCR, *Escherichia coli* strain EDL933 DNA was added to each sample as an internal inhibition control, using primers and probes specific for the *E. coli eae* gene [13].

### 2.6. Validation of the Results by Real-Time PCR, Virus Isolation, and Genome Sequencing

When cDNA of pools of abdomens were detected positive for viruses, the cDNAs of RBP (head/thorax) of individual mosquitoes composing each pool were screened by real-time PCRs on a LightCycler^®^ 480 (LC480) (Roche Applied Science, Penzberg, Germany). Real-time PCR assay targeting the virus of interest (see primers/probe sets in Appendix A) was performed in a final volume of 12 µL using the LightCycler^®^ 480 Probe Master Mix 1X (Roche Applied Science, Germany), with primers and probes at 200 nM and 2 µL of control DNA. Thermal cycling conditions were as follows: 95 °C for 5 min, 45 cycles at 95 °C for 10 s, and 60 °C for 15 s, and one final cooling cycle at 40 °C for 10 s.

When a positive sample was confirmed, virus isolation was attempted in Vero and C6/36 cells. Then, total RNA was extracted using the Nucleospin RNA II extract kit (Macherey-Nagel, Hoerdt, France) following the manufacturer instructions and full genome sequencing was attempted. For ZIKV, 12 overlapping amplicons were produced using the reverse transcriptase Platinum Taq High Fidelity polymerase enzyme (Thermo Fisher Scientific, Waltham, MA, USA) and specific primers (Appendix A). PCR products were pooled in equimolar proportions. After Qubit quantification using Qubit^®^ dsDNA HS Assay Kit and Qubit 2.0 fluorometer (Thermo Fisher Scientific, Waltham, MA, USA), amplicons were fragmented (sonication) into fragments of 200 bp long. Libraries were built adding barcode for sample identification, and primers to fragmented DNA using AB Library Builder System (Thermo Fisher Scientific, Waltham, MA, USA). To pool the barcoded samples equimolarly, a quantification step by the 2100 Bioanalyzer instrument (Agilent Technologies, Santa Clara, California, USA) was performed. An emulsion PCR of the pools and loading on 520 chip was done using the automated Ion Chef instrument (Thermo Fisher Scientific, Waltham, MA, USA). Sequencing was performed using the S5 Ion torrent technology (Thermo Fisher Scientific, Waltham, MA, USA) following manufacturer’s instructions. Consensus sequence was obtained after removing the 30 first and last nucleotides of each read, trimming reads depending on quality (reads with quality over >99%) and length (reads over 100 pb were kept), and mapping them on a reference (KY415987, most similar sequence after Blastn) using CLC genomics workbench software 11.0.1 (Qiagen, Hilden, Germany). A de novo contig was also produced to ensure that the consensus sequence was not affected by the reference sequence. 

## 3. Results

One hundred and forty-nine primer/probe sets were designed to detect 64 MBV (Appendix A). Among them, 95 sets of primers/probe specifically identified their corresponding positive control samples (37 viral RNA) via Taqman RT-real-time PCRs or Taqman real-time PCRs on a LightCycler 480 apparatus. Resulting C_t_ values varied from 8 to 42 depending on sample type and nucleic acid concentration. Unfortunately, 54 designs were not tested due to the lack of RNA positive control.

To avoid sensitivity problems, cDNA pre-amplification was included in the assay. This step enabled detection of all positive controls (95 primer/probe sets tested on 37 viral RNAs) via Taqman real-time PCRs on a LC480 apparatus. The specificity of each primers/probe set was then evaluated using 37 MBV positive controls on the BioMark™ system (Figure 1A,B). Results demonstrated high specificity for each primer/probe set after pre-amplification (Figure 1A,B). Indeed, 91 assays (among the 149 developed) were only positive for their corresponding positive controls. Four designs demonstrated cross-reactivity with a virus from the same species or genus: DENV-1 assay amplified also DENV-2, DENV-2 assay cross-reacted with DENV-3 and DENV-4, DENV-4 cross-reacted with DENV-3, and one WNV assay amplified Usutu virus (USUV). Specificity of 54 assays was not fully tested in the absence of their respective positive controls. Nevertheless, those designs did not show any cross-reaction with RNA positive controls from other viruses.

### 3.1. Laboratory-Infected Mosquitoes

Forty-seven batches each containing three infected mosquitoes were screened with the high-throughput technique developed. The system was able to identify the six viruses present in different mosquitoes (Figure 2). Indeed, seven batches were infected by DENV-1, four by DENV-3, four by DENV-4, 3 by CHIKV, five by WNV, 13 batches by ZIKV, and 10 batches were coinfected by CHIKV and DENV-2. As for the specificity test, DENV-1, DENV-2, and DENV-3 assays demonstrated cross-reactions. 

### 3.2. Field-Collected Mosquitoes from Endemic and Epidemic Areas

A total of 17,958 field-collected mosquitoes in six countries from the African, American, and Asian continents were screened for arbovirus.

### 3.3. Endemic Areas

#### 3.3.1. Senegal

In Senegal, 934 arthropods including 6 sandflies and 928 mosquitoes (25 males and 909 females) from 21 species and five genera (detailed in Table 1), were collected in the Kedougou area (Southeastern Senegal) from August to November 2017. Moreover, 402 larvae were also collected in the same area from August 2017 to January 2018 and reared until adult emergence in insectarium (188 males and 214 females obtained). Mosquitoes were grouped by species and sex; 231 and 112 pools were respectively analyzed for MBVs. YFV was detected in one pool of 20 females of *Aedes furcifer* and was confirmed in head/thorax from 1 *Aedes furcifer* female by RT-real-time PCR. Virus isolation was attempted but without any success.

#### 3.3.2. Cambodia

In Cambodia, 492 mosquitoes (73 males and 419 females) from 28 species and 5 genera (detailed in Table 2), were collected in one area at two periods, the dry season in May 2019 and the rainy season in November 2018. Mosquitoes were grouped by species and sex into 109 pools and were analyzed for MBVs. No virus was detected (Table 2).

### 3.4. Endemic/Epidemic area, Brazil

In Brazil, 7705 mosquitoes (889 males and 6816 females) belonging to 22 species and 15 genera (detailed in Table 3) were collected in 15 areas from January 2016 to May 2017. Mosquitoes were then grouped into 647 pools and were analyzed for MBVs. In total, three different viruses (YFV, CHIKV, and Trivittatus virus (TVTV)) were preliminary detected in six pools (in four, one, and one, respectively). Only the presence of YFV was confirmed in the head/thorax from individual mosquitoes, from three species (*Ae. scapularis, Ae. taeniorhynchus*, and *Hg. leucocelaenus*) by RT real-time PCR corresponding to YFV strains currently circulating in Brazil (37). Attempts to isolate the virus were made but remained unsuccessful.

### 3.5. Epidemic Areas

#### 3.5.1. Guadeloupe

In Guadeloupe, 150 mosquito pools corresponding to 2173 mosquitoes (884 males and 1289 females) from five species (*Ae. aegypti*, *Culex quinquefasciatus*, *Anopheles albimanus*, *Cx. bisulcatus*, *Cx. nigripalpus*, 54 *Culex.* spp.) collected from May to June 2016 were screened for 64 MBVs. ZIKV was found in two pools of *Cx. quinquefasciatus* females and nine pools of *Ae. aegypti* (eight pools of females and one pool of males) (Table 4). ZIKV was detected only in the head/thorax of individual *Ae. aegypti* females from the eight positive pools by a RT-real-time PCR. Virus was isolated on Vero cells and full genome sequencing identified the Asian genotype (GenBank Accession Numbers: MN185324, MN185325, MN185327, MN185329, MN185330, MN185331, MN185332).

#### 3.5.2. French Guiana

In French Guiana, 3942 mosquitoes (1098 males and 2844 females) from seven species (*Ae. aegypti, Ae. scapularis, Ae. taeniorhynchus*, *Cx. quinquefasciatus, Ma. titillans, Cq. venezualensis,* and *Cq. albicosta*) were collected in the Cayenne area from June to August 2016 and grouped into 248 pools to be screened. Three pools of *Ae. aegypti* and one pool of *Cx. quinquefasciatus* were detected positive for ZIKV (Table 5). After screening individual head/thorax from those pools, only pools of *Ae. aegypti* were confirmed positive. ZIKV was isolated and fully sequenced; it belonged to the Asian genotype (GenBank Accession numbers: MN185326 and MN185328).

#### 3.5.3. Suriname

In Suriname, from March to May 2017, four species/genus of mosquitoes (2256 *Ae. aegypti*, 29 *Culex* spp., 5 *Haemogogus* spp., 20 undetermined species) representing 2310 adults, were grouped into 77 pools and screened. No virus was detected (Table 6).

## 4. Discussion

In this study, we developed a new high-throughput virus-detection assay based on microfluidic PCRs able to detect 64 MBVs in mosquitoes. Only four primer sets demonstrated cross-reactivity with viruses from the same genus or serotype. Moreover, specificity of 54 assays was not fully tested in the absence of their respective positive controls. Nevertheless, these designs did not show any cross-reaction with RNA positive controls from other viruses. Subsequently, we used this newly developed assay to perform a large epidemiological survey screening in six countries/territories during the last Zika pandemic. This new method has allowed the detection of (i) three human infecting arboviruses, ZIKV, YFV, and CHIKV, in mosquitoes and (ii) other unexpected viruses such as TVTV.

The efficiency of our tool was the first requirement; we used artificially infected mosquitoes to detect different viruses (DENV1-4, CHIKV, WNV, ZIKV) offered in single and dual infections. Our assays can target DENV with however cross-reactions between serotypes. Caused by one of the four serotypes (DENV1-4), dengue is the most important arboviral disease worldwide [14]. It is widely accepted that a subsequent infection with a second serotype can produce more severe symptoms [15]. This situation becomes challenging when multiple serotypes co-circulate [16]. Mosquitoes co-infected with different DENV serotypes are occasionally detected [17]. *Aedes aegypti* and *Ae. albopictus* are urban vectors of DENV responsible for most epidemic outbreaks in Asia, Latin America, the Caribbean, and Pacific islands [14]. Co-infected *Aedes* mosquitoes are capable of transmitting multiple arboviruses during one bite [9]. Dual DENV detections in mosquitoes may be a sign of co-circulation of DENV and then may help in predicting co-infections in humans. Diagnosis of dengue infections cannot be based on clinical symptoms as dengue disease shares common symptoms with other arboviral diseases [18]. To discriminate dengue serotypes, viral isolation and viral RNA detection remain the gold standard methods but should be performed during patient viremia (within five days after the onset of fever). Less constraining and costly, mass viral screening of mosquitoes in surveillance and epidemic contexts can be an advantageous substitute.

In the same way, WNV assays cross-reacted with the phylogenetically-related USUV. WNV is a flavivirus responsible of neuro-invasive disease in Europe and North America [19]. Diagnosis of WNV infection remains challenging and human cases are usually underestimated. On the other hand, USUV has spread over Europe during the last 20 years causing bird mortalities and some rare human cases [20]. Human infections are rare and often asymptomatic, and neurological disorders can be described [20]. While WNV has circulated in Europe since the 1960s, USUV shares the same geographical distribution and also the same vectors, *Culex pipiens*. Our tool did not succeed in distinguishing the two viruses, and therefore it needs more improvements.

By screening 17,958 mosquitoes collected in six countries/territories for 64 different MBVs, we succeeded in detecting ZIKV, YFV, CHIKV, and TVTV in mosquitoes.

The Zika outbreak was unexpected; the first human cases outside endemic regions in Africa were reported in Yap island in 2007 where the outbreak was poorly publicized despite the two third of the population affected [21]. Few years later, ZIKV hit French Polynesia [22] where the first notification of severe symptoms associated to ZIKV infections were done, Guillain–Barre syndrome [23] and microcephaly in new-born [24]. After ZIKV reached the American continent in 2015 [25], phylogenetic analysis indicated that the circulating ZIKV belonged to the Asian clade [26,27]. Our tool was able to detect ZIKV in pools of abdomen of *Ae. aegypti* and *Cx. pipiens* from Guadeloupe and French Guiana. However, when analyzing disseminated viral particles in head and thorax, only *Ae. aegypti* was found infected, corroborating the main role of this species in ZIKV transmission and limiting *Cx. quinquefasciatus* and *Cx. pipiens* as a vector [28]. It is widely admitted that viral dissemination beyond the midgut can be a clue attesting the mosquito susceptibility to a virus. However, viral dissemination in mosquitoes depends on mosquito collection date; it increases over time after the infectious blood meal [3]. In our study, it was not possible to have information on the physiological age of mosquitoes and when they were infected.

Our mass screening tool has detected YFV in five mosquito species: *Aedes scapularis, Aedes taeniorhynchus, Haemagogus janthinomys, Haemagogus leucocelaenus*, and *Sabethes chloropterus*. The species *Hg. janthinomys* and *Hg. leucocelaenus* are considered as the main vectors of YFV in Brazil [29,30] while *Aedes scapularis, Aedes taeniorhynchus*, and *Sabethes chloropterus* only play a secondary role [31]. Other viruses preliminarily detected in Brazilian mosquitoes were TVTV and CHIKV. While CHIKV continues to cause sporadic cases in Brazil after the massive outbreak in 2015, TVTV was first isolated from *Aedes trivittatus* in USA in 1948, and has never been detected outside North America where it is mainly distributed [32]. Consequences of TVTV infections on humans remain unknown [33]. Nevertheless, the presence of CHIKV and TVTV in tested mosquitoes was not confirmed. 

This study demonstrates the feasibility of high-throughput screening methods to detect diverse MBVs in field-collected mosquitoes. Performing 9216 real-time PCRs in one run took four hours, and the cost was around $10 per reaction from sample homogenization to virus detection by real-time PCR [10,11]. Nevertheless, the instrument is costly and requires some specific conditions of use. It is therefore recommended to identify focal points where this technology could be developed and to improve the conditions for transporting biological samples from the field to allow an optimal viral screening. Another main advantage of our tool is the adaptability of the system by adding new sets of primers and probes targeting newly emergent viruses in contrast to arrays with fixed panels of probes. Indeed, because the number of YFV cases was unusually high since January 2016 [34], we added specific detections of YFV strains circulating in South America to screen field-collected mosquitoes from Brazil, French Guiana, Suriname, and Guadeloupe. In conclusion, our method designed to specifically identify MBVs in mosquitoes can be used to screen other types of samples such as human and/or animal blood or organs [35]. We demonstrated the usefulness of this new screening method, which represents a powerful, cost-effective, and rapid system to track MBVs all around the world and could be easily customized to any viral emergence.

## Figures and Tables

**Figure 1 viruses-11-00904-f001:**
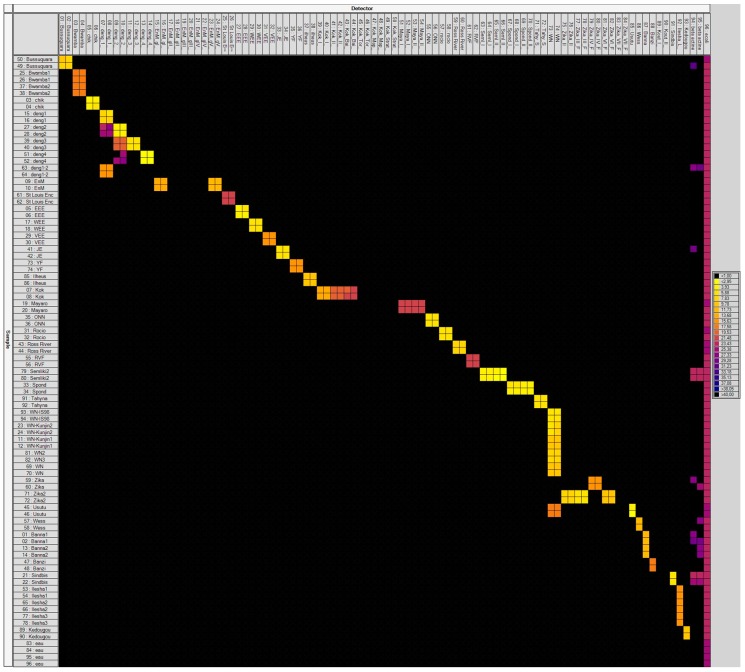
BioMark™ dynamic array system specificity test (96.96 chip). Specificity of primers/probe sets from the Appendix A are presented in two figures, (**A**,**B**). Fifty-one sets of primers/probe targeting viruses are presented in (A) and 94 sets in (B), some of them are present into both figures. Each square corresponds to a single real-time PCR reaction, where rows indicate the pathogen in the positive control and columns represent the targets of each primer/probe set. C_t_ values for each reaction are indicated in color; the corresponding color scale is presented in the legend on the right. The darkest shade of blue and black squares are considered as negative reactions with C_t_ > 30. chik: Chikungunya; deng: Dengue; EnM: Murray Encephalitis; St Louis Enc: Saint Louis Encephalitis; EEE: Eastern Equine Encephalitis; WEE: Western Equine Encephalitis; VEE: Venezuelan Equine Encephalitis; JE: Japanese Encephalitis; YF: Yellow fever; Kok: Kokobera group; Maya: Mayaro; ONN: O’nyong nyong; RVF: Rift Valley Fever; Seml: Semliki forest; Spond: Spondweni; Tahy: Tahyna; WN: West Nile; Wess: Wesselsbron; Kout: Koutango; Guama: Guama Group; Calif Enc: California Encephalitis; GpC: C Group; Key: Keystone; Snow: Snowshoe hare.

**Figure 2 viruses-11-00904-f002:**
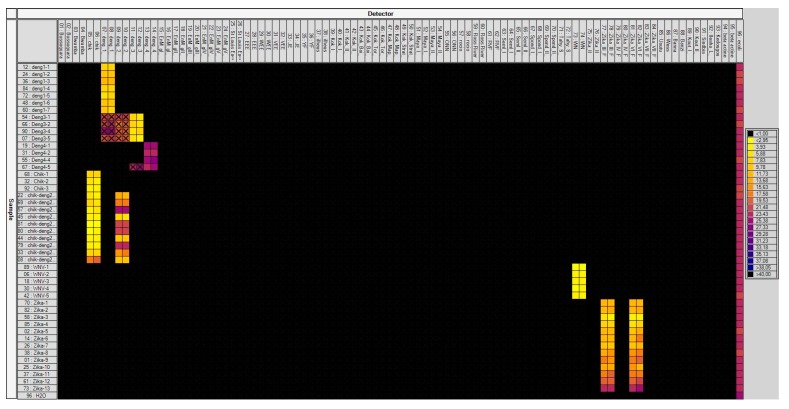
Screening of artificially infected mosquitoes through the BioMark™ dynamic array system developed (96.96 chip). Each square corresponds to a single real-time PCR reaction, where rows indicate the batches of mosquitoes tested and columns represent the targets of each primer/probe set. Crosses indicate cross-reaction of assays. C_t_ values for each reaction are indicated in color; the corresponding color scale is presented in the legend on the right. The darkest shade of blue and black squares are considered as negative reactions with C_t_ > 30. chik: Chikungunya; deng: Dengue; EnM: Murray Encephalitis; St Louis Enc: Saint Louis Encephalitis; EEE: Eastern Equine Encephalitis virus; WEE: Western Equine Encephalitis; VEE: Venezuelan Equine Encephalitis; JE: Japanese Encephalitis; YF: Yellow fever; Kok: Kokobera grou; Maya: Mayaro; ONN: O’nyong nyong; RVF: Rift Valley Fever; Seml: Semliki fores; Spond: Spondweni; Tahy: Tahyna; WN: West Nile; Wess: Wesselsbron; Kout: Koutango.

**Table 1 viruses-11-00904-t001:** Mosquito and sandflies species, number of mosquitoes collected, and number of pools analyzed in Senegal.

Stage Collected	Collection SITE	GPS Coordinates	Urban Rural Sylvatic	Mosquito Species	Number of Arthropods Screened	Virus Detected through Microfluidic System	Type of Confirmation Performed
Adult	Baraboye	12°41′11.2″ N/12°24′39.2″ W	Rural	*Ae. furcifer, Ae. dalzieli, Ae. taylori, Ae. vittatus, An. coustani, An. funestus, Ma. uniformis,* sandflies	97	-	-
	Ngari	12°38′07.3″ N/12°14′59.5″ W	Rural	*Ae. furcifer, Ae. dalzieli, Ae. aegypti, Ae. argenteopunctatus, Ae. hirsutus, Ae. mcintoshi, Cx. quinquefasciatus, Ae. vittatus, An. coustani, An. funestus,*	96	-	-
	Silling	12°32′36.5″ N/12°16′18.7″ W	Rural	*Ae. luteocephalus, An. gambiae*	3	-	-
	Tenkoto	12°40′23.1″ N/12°16′37.1″ W	Rural	*Ae. furcifer*	5	-	-
	Velingara	12°27′33.9″ N/12°03′17.3″ W	Rural	*Ae. aegypti, Ae. dalzieli, Ae. furcifer, Ae. luteocephalus, Ae. vittatus, An. coustani*	43	-	-
	Kedougou (C1F)	12°39′42.1″ N/12°16′05.2″ W	Sylvatic	*Ae. aegypti, Ae. furcifer, Ae. luteocephalus, Ae. taylori, Ae. unilineatus, Ae. vittatus, An. coustani, An. funestus, An. nili, Cx. perfuscus, Ma. africana, Ma. uniformis*	151	-	-
	Kedougou (D1F)	12°36′43.9″ N/12°14′50.7″ W	Sylvatic	*Ae. aegypti, Ae. africanus, Ae. dalzieli, Ae. furcifer, Ae. luteocephalus, Ae. taylori, Ae. unilineatus, Ae. vittatus, An. coustani, An. funestus, An. nili, Cx. annulioris, Cx. bitaeniorhynchus, Cx. poicilipes, Cx. perfuscus, Ma. africana, Ma. uniformis*	278	YFV *	Literature
	Kedougou (E2F)	12°29′21.2″ N/12°06′06.2″ W	Sylvatic	*Ae. aegypti, Ae. africanus, Ae. argenteopunctatus, Ae. dalzieli, Ae. furcifer, Ae. luteocephalus, Ae. taylori, Ae. unilineatus, Ae. vittatus, An. coustani, An. funestus, Cx. poicilipes, Ma. uniformis*	261	-	-
**Total**	**8**				**934**		
Larvae	Dalaba	12°33′25.6″ N/12°10′41.0″ W	Rural	*Ae. aegypti*	24	-	-
	Ngari	12°38′07.3″ N/12°14′59.5″ W	Rural	*Ae. aegypti, Ae. vittatus*	18	-	-
	Kedougou (D1F)	12°36′43.9″ N/12°14′50.7″ W	Sylvatic	*Ae. aegypti, Ae. bromeliae, Ae. furcifer, Ae. longipalpis, Ae. luteocephalus, Ae. taylori, Ae. unilineatus, Ae. vittatus*	237	-	-
	Kedougou (E2F)	12°29′21.2″ N/12°06′06.2″ W	Sylvatic	*Ae. aegypti, Ae. bromeliae, Ae. longipalpis, Ae. luteocephalus, Ae. neoafricanus, Ae. taylori, Ae. unilineatus, Ae. vittatus, Er. chrysogaster*	123	-	-
**Total**	**4**				**402**		

*Ae., Aedes; An., Anopheles; Cx., Culex; Er., Eretmapodites; Ma., Mansonia; YFV, Yellow fever virus*. (-): no virus detected and no confirmation performed. * YFV detected in one pool of *Ae. scapularis*, one pool of *Ae. furcifer* and confirmed in one female of this pool.

**Table 2 viruses-11-00904-t002:** Mosquito species, number of mosquitoes collected, and number of pools analyzed in Cambodia.

Collection Site	GPS Coordinates	Urban Rural Sylvatic	Mosquito Species	Number of Mosquitoes Screened	Virus Detected through Microfluidic System	Type of Confirmation Performed
Mondulkiri, Cambodia	12°10′28.4″/106°53′40.9″	Sylvatic	*Ae. aegypti, Ae. albopictus, Ae. gardnerri imitator, Ae. prominens, An. barbirostris, An. indefinitus, An. jamesii, An. kochi, An. mimulus complex, An. philippinensis, An. roperi, An. umbrosus, An. vegus, Ar. annulipalpis, Ar. dolichocephalus, Ar. flavus, Ar. foliatus/kuchingensis, Ar. moultoni, Ar. subalbatus, Ar. theobaldi, Cx. bitaeniorhynchus, Cx. brevipalpis, Cx. fuscocephala, Cx. perplexus/whitei, Cx. sitiens, Cx. vishnui complex, Hz. catesi, Hz. demeilloni*	492	-	-

Ae., Aedes; An., Anopheles; Ar., Armigeres; Cx., Culex; Hz., Heizmannia. (-): no virus detected and no confirmation performed.

**Table 3 viruses-11-00904-t003:** Mosquito species, number of mosquitoes collected, and viruses detected in Brazil.

Collection Site	GPS Coordinates	Urban Rural Sylvatic	Mosquito Species	Number of Mosquitoes Screened	Virus Detected through Microfluidic System	Type of Confirmation Performed
Belo Horizonte	19°51′59.29″ S/44° 0′43.51″ W	Urban Forest	*Ae. albopictus, Ae. aegypti, Cx. quinquefasciatus, Sa. albiprivus*	17	-	-
Casimiro de Abreu	22°26′33.31″ S/42°12′30.34″ W	Sylvatic	*Ae. scapularis*	24	-	-
Domingos Martins	20°17′12.48″ S/40°50′14.35″ W	Sylvatic	*Ae. albopictus*	7	-	-
Goiânia	6°40′16.32″ S/49°22′49.93″ W	Urban Forest	*Aedeomiya, Ae. aegypti, Ae. albopictus, Aedes sp., Coquillettidia sp., Culex sp., Hg. leucocelaenus, Limatus sp., Mansonia sp., Orthopodomyia sp., Psorophora sp., Sabethes sp., Wyeomyia sp.*	689	-	-
Guapimirim	22°28′56.31″ S/42°59′26.36″ W	Sylvatic	*Ru. frontosa*	10	-	-
Macaé	22°18′17.54″ S/42° 0′8.80″ W	Sylvatic	*Ae. scapularis, Wyeomyia sp.*	25	-	-
Manaus	3°00′12.78″ S/59°55′37.86″ W	Urban Forest	*Ae. aegypti, Ae. albopictus, Aedes sp., Culex sp., Hg. leucocelaenus, Limatus sp., Orthopodomyia sp., Psorophora sp., Sabethes sp., Trichoposopum sp., Uranotenia sp., Wyeomyia sp.*	3939	CHIKV *	Isolation, conventional and real-time PCR failed to confirm the result
Maricá	22°55′24.44″ S/42°42′27.88″ W	Sylvatic	*Ae. aegypti, Ae. albopictus, Ae. scapularis, Ae. taeniorhynchus, Culex sp., Cx. nigripalpus, Hg. janthinomys, Hg. leucocelaenus, Li. durhamii, Ru. humboldti*	198	YFV ^§^	YFV confirmed by PCR
Miguel Pereira	22°29′3.21″ S/43°18′15.98″ W	Sylvatic	*Sh. fluviatilis*	12	-	_
Nova Friburgo	22°24′46.35″ S/42°18′58.57″ W	Sylvatic	*Ru. humboldti*	2	CHIKV ^£^	Isolation, conventional and real-time PCR failed to confirm the result
Queluz	22°41′52.33″ S/44°43′43.91″ W	Sylvatic	*Wy. pilicauda, Wy. confusa*	6	-	-
Rio de Janeiro	22°52′45.7″ S/43°18′10.0″ W	Urban	*Ae. aegypti, Ae. albopictus, Cx. quinquefasciatus*	261	-	-
	22°56′6.57″ S/43°26′42.19″ W	Urban Forest	*Ae. aegypti, Ae. albopictus, Aedes sp., Coquillettidia sp., Culex sp., Hg. leucocelaenus, Mansonia sp., Psorophora sp., Runchomyia sp., Sabethes sp., Trichoposopum sp., Wyeomyia sp.*	2447	TVTV ^$^	Isolation, conventional and real-time PCR failed to confirm the result
Serra	20° 6′46.89″ S/40°11′12.53″ W	Sylvatic	*Ae. albopictus, Cx. quinquefasciatus*	26	-	-
Simonésia	19°55′12.06″ S/41°54′20.23″ W	Sylvatic	*Ae. albopictus, Cq. venezuelensis, Hg. janthinomys, Hg. leucocelaenus, Sa. albiprivus*	41	-	-
Teresópolis	22°26′58.56″ S/42°59′5.43″ W	Sylvatic	*Ru. frontosa*	1	-	-
**Total 15**				**7705**		

Ae., Aedes; Cq., Coquillettidia; Cx., Culex; Hg., Haemagogus; Li., Limatus; Ru., Runchomyia; Sa., Sabethes; Sh., Shannoniana; Wy., Wyeomyia; CHIKV, Chikungunya virus; YFV, Yellow fever virus; TVTV, Trivittatus virus. (-): no virus detected and no confirmation performed. * CHIKV detected in one pool of Cx. erraticus; ^§^ YFV detected in one pool of Ae. scapularis, one pool of Ae. taeniorhynchus, one pool of Hg. leucocelaenus; ^£^ CHIKV detected in one pool of Ru. humboldti; ^$^ TVTV detected in one pool of Cx. nigripalpus.

**Table 4 viruses-11-00904-t004:** Mosquito species, number of mosquitoes collected, and virus detected in Guadeloupe.

Collection Site	GPS Coordinates	Urban RuralSylvatic	Mosquito Species	Number of Mosquitoes Screened	Virus Detected through Microfluidic System	Type of Confirmation Performed
Gosier	16° 12′ 21.229″ N/61° 29′ 31.438″ W	Urban	*Ae. aegypti, Cx. quinquefasciatus, An. albimanus*	399	-	-
Deshaies	16° 18′ 24.973″ N/61° 47′ 39.556″ W	Urban/Periurban	*Ae. aegypti, Cx. quinquefasciatus, An. albimanus, Cx. bisulcatus, De. magnus*	306	ZIKV *	Confirmed by real-time PCR on head-thorax of individual mosquitoes, isolation of the virus, and full genome sequencing
Petit Bourg	16° 11′ 29.476″ N/61° 35′ 25.753″ W	Urban/Periurban	*Ae. aegypti, Cx. quinquefasciatus, Cx. nigripalpus, Culex sp.*	422	ZIKV *	Confirmed by real-time PCR on head-thorax of individual mosquitoes, isolation of the virus, and full genome sequencing
Le Moule	16° 19′ 52.342″ N/61° 20′ 37.41″ W	Urban/Periurban	*Ae. aegypti, Cx. quinquefasciatus, Culex sp.*	202	ZIKV *	Confirmed by real-time PCR on head-thorax of individual mosquitoes, isolation of the virus, and full genome sequencing
Saint François	16° 15′ 5.141″ N/61° 16′ 26.825″ W	Urban	*Ae. aegypti, Cx. quinquefasciatus*	356	ZIKV *	Confirmed by real-time PCR on head-thorax of individual mosquitoes, isolation of the virus, and full genome sequencing
Sainte Anne	16° 13′ 31.613″ N/61° 23′ 9.377″ W	Urban/Periurban	*Ae. aegypti, Cx. quinquefasciatus, Cx. nigripalpus, Culex sp.*	325	ZIKV *	Confirmed by real-time PCR on head-thorax of individual mosquitoes, isolation of the virus, and full genome sequencing
Baie Mahault	16° 16′ 3.979″ N/61° 35′ 13.337″ W	Urban	*Ae. aegypti, Cx. quinquefasciatus*	19	-	-
Le Lamentin	16° 16′ 17.36″ N/61° 37′ 59.754″ W	Urban/Periurban	*Ae. aegypti, Cx. quinquefasciatus*	27	-	-
Goyave	16° 7′ 26.447″ N/61° 34′ 40.253″ W	Urban/Periurban	*Ae. aegypti, Cx. quinquefasciatus, Culex sp.*	86	-	-
Morne-à-L′eau	16° 19′ 53.832″ N/61° 27′ 25.855″ W	Urban	*Ae. aegypti, Cx. quinquefasciatus*	19	-	-
Pointe-à-Pître	16° 14′ 54.499″ N/61° 32′ 18.888″ W	Urban	*Ae. aegypti, Cx. quinquefasciatus*	5	-	-
Saint-Claude	16° 1′ 36.077″ N/61° 42′ 6.703″ W	Urban/Periurban	*Ae. aegypti*	1	-	-
Petit Canal	16° 22′ 49.03″ N/61° 29′ 14.384″ W	Urban/Periurban	*Culex sp.*	6	-	-
**Total 13**				**2173**		

*Ae., Aedes; An., Anopheles; Cx., Culex; De., Deinocerites*. (-): no virus detected and no confirmation performed. * ZIKV detected in 2 pools of *Cx. quinquefasciatus* and 9 pools of *Ae. aegypti*, and confirmed in 9 females *Ae. aegypti* (one female per pool).

**Table 5 viruses-11-00904-t005:** Mosquito species, number of mosquitoes collected, and virus detected in French Guiana.

Collection Site	GPS Coordinates	Urban Rural Sylvatic	Mosquito Species	Number of Mosquitoes Screened	Virus Detected through Microfluidic System	Type of Confirmation Performed
Cayenne	4°55′53.08″ N/52°18′55.99″ W	Urban	*Ae. aegypti, Ae. scapularis, Cx. quinquefasciatus, Ma. titillans, Cq. venezualensis, Cq. albicosta*	1928	ZIKV *	Confirmed by real-time PCR on head-thorax of individual mosquitoes, isolation of the virus, and full genome sequencing
Remire-Montjoly	4°53′34.01″ N/52°16′34.32″ W	Urban/Periurban	*Ae. aegypti, Ae. scapularis, Cx. quinquefasciatus, Ma. titillans, Cq. venezualensis*	1078	_	-
Matoury	4°50′52.22″ N/52°19′41.58″ W	Urban/Periurban	*Ae. aegypti, Ae. taeniorhynchus, Cx. quinquefasciatus, Ma. titillans, Cq. venezualensis, Cq. albicosta*	936	ZIKV *	Confirmed by real-time PCR on head-thorax of individual mosquitoes, isolation of the virus, and full genome sequencing
**Total 3**				**3942**		

*Ae., Aedes; Cq., Coquillettidia; Cx., Culex; Ma., Mansonia.* (-): no virus detected and no confirmation performed. * ZIKV in 1 pools of *Cx. quinquefasciatus* and 3 pools of *Ae. aegypti*, and confirmed in 3 females *Ae. aegypti* (one female per pool).

**Table 6 viruses-11-00904-t006:** Mosquito species, number of mosquitoes collected, and virus detected in Suriname.

Collection Site	GPS	Urban Rural Sylvatic	Mosquito Species	Number of Mosquitoes Screened	Virus Detected through Microfluidic System	Type of Confirmation Performed
Paramaribo	5°51′54.2″ N/55°11′33.4″ W	Urban (Paramaribo)	Undetermined	4	-	-
Roti shop	5°51′59.616″ N/55°6′20.952 W	Urban (Paramaribo)	*Ae. aegypti*	68	-	-
Car mechanic	5°50′35.8″ N 55°06′56.7″ W	Urban (Paramaribo)	*Ae. aegypti*	96	-	-
Kwikfit car mechanic	5°50′38.1″ N 55°07′23.3″ W	Urban (Paramaribo)	*Ae. aegypti*	567	-	-
Family home	5°50′33.3″ N 55°07′17.5″ W	Urban (Paramaribo)	*Ae. aegypti*	296	-	-
Outpatient clinic	5°50′30.3″ N/55°7′8.615″ W	Urban (Paramaribo)	*Ae. aegypti*	103	-	-
Chi min restaurant	5°49′54.408″ N/55°8′24.683 W	Urban (Paramaribo)	*Ae. aegypti*	78	-	-
Albertine retirement home	5°48′49.572″ N/55°11′27.6″ W	Urban (Paramaribo)	*Ae. aegypti*	661	-	-
Medisch Opvoedkundig Bureau (MOB)	5°49′43.212″ N/55°10′41.375″ W	Urban (Paramaribo)	*Ae. aegypti*	376	-	-
Brownsweg	5°0′57.384″ N/55°10′2.172″ W	Rural/Sylvatic	*Ae. aegypti*, *Culex* sp., Undetermined	21	-	-
Brownsberg	4°56′36.24″ N/55°10′6.6″ W	Rural/Sylvatic	*Ae. aegypti*, *Culex* sp., *Haemagogus* sp., Undetermined	40	-	-
**Total 11**				**2310**		

*Ae., Aedes; Cx., Culex; Hg., Haemagogus.* (-): no virus detected and no confirmation performed.

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
