# Peer review of "A New High-Throughput Tool to Screen Mosquito-Borne Viruses in Zika Virus Endemic/Epidemic Areas"

_viruses, 2019, doi:10.3390/v11100904_

Round 1
Reviewer 1 Report
In order to detect emerging of arboviral infection in human populations, Moutailler and co-workers sought to establish a high throughput system able to detect a wide range of arbovirus in a large population of field-collected mosquitoes. To this aim they established a real-time microfluidic PCR system. The approach was first validated on artificially infected mosquitoes and then deployed in different contexts were endemic infections of major human arboviruses were present.
First, the authors tested and validated a library of specific primers and probes to detect 64 different mosquito borne viruses. The authors tested the ability of the primers to amplify the respective positive controls and their specificity crossing a set of primers with all the other available controls. This allowed to assess the high specificity of the assay, since only four designs exhibited cross-reactivity.
The manuscript is well written, the conclusions are supported by the data. The authors have developed an useful tool that could allow to screen large number of samples in a cost-effective way. Moreover, the primers/probe library developed by the authors will be useful to other researcher attempting to detect different mosquito borne viruses thanks to their proven specificity.
I have only few minor comments that should be addressed to make the manuscript more clear.
Figure 1 is of difficult reading. The columns and rows labels are not clear, while it is still possible to see labels in fig1A, fig1B is not readable. Moreover the black background does not allow an easy navigation of the table. I would suggest using a grid to facilitate the reading. Moreover, it is not clear to me what is the difference between Fig1A and Fig1B, since several identical set of samples/detectors are reported on both figures. The author should clarify it in the text of in the figure legend.
Same problems affect figure 2. Each row corresponds to a set of infected mosquitoes but it is not clear with which virus they were infected since the labels are not specified anywhere in the figure legend. For instance, the authors used the wording “deng” to indicate primers that would amplify dengue genome, but use only the letter D to label mosquitoes infected with dengue. I my opinion this generate confusion. The authors should unify their labels using the same identifier for primers/probe and target virus. Moreover, two digits numbers are present in the labels of the rows. What do they indicate? If they are part of an internal code, they are not useful for the reader.
In the discussion the authors claimed that they are able to detect 64 MBVs, however positive controls used for validation were available only for 37 of these MBVs. Thus the authors have validated the detection of 37 MBVs and not 64 as they state in the conclusions.
In the method section, the authors should state how the batches of infected mosquitoes were prepared.
Author Response
"A new high-throughput tool to screen mosquito-borne viruses in Zika virus endemic/epidemic areas " (viruses-603136) by Moutailler et al.
Dear Editor,
On behalf of my co-authors, I would like to thank the reviewers for the critical reading of our manuscript. We have considered their requests and made most changes suggested. Please, find below point-by-point answers to reviewers, indicating which changes have been made, and where they have been inserted in the text (highlighted in yellow in the revised version).
Sincerely yours,
Sara Moutailler
Anna-Bella Failloux
Answers to Reviewer 1
Point #1: Figure 1 is of difficult reading. The columns and rows labels are not clear, while it is still possible to see labels in fig1A, fig1B is not readable. Moreover the black background does not allow an easy navigation of the table. I would suggest using a grid to facilitate the reading. Moreover, it is not clear to me what is the difference between Fig1A and Fig1B, since several identical set of samples/detectors are reported on both figures. The author should clarify it in the text of in the figure legend.
We have modified the columns and rows labels of the figures 1A/B and 2 as suggested allowing a clearer reading of figures. We could only run 94 sets of primers/probe targeting viruses per chip. So to test all the 149 sets developed in our system, two different chips have been run corresponding to the figures 1 A and 1B. And because we could not have empty place in a chip, some designs have been run in double in the chips 1A and 1B. An explanation has been added in the figure legend: “51 sets of primers/probe targeting viruses are presented in Fig. 1A and 94 sets in Fig. 1B, some of them are present into both figures.”
Point #2: Same problems affect figure 2. Each row corresponds to a set of infected mosquitoes but it is not clear with which virus they were infected since the labels are not specified anywhere in the figure legend. For instance, the authors used the wording “deng” to indicate primers that would amplify dengue genome, but use only the letter D to label mosquitoes infected with dengue. I my opinion this generate confusion. The authors should unify their labels using the same identifier for primers/probe and target virus.
We have modified the columns and rows labels of the figures 1A/B and 2 as suggested to allow a clearer reading of figures.
Point #3: Moreover, two digits numbers are present in the labels of the rows. What do they indicate? If they are part of an internal code, they are not useful for the reader.
We thanks the reviewer for this comment. Indeed, these digits numbers are internal code. We have modified rows labels as suggested.
Point #4: In the discussion the authors claimed that they are able to detect 64 MBVs, however positive controls used for validation were available only for 37 of these MBVs. Thus the authors have validated the detection of 37 MBVs and not 64 as they state in the conclusions.
We have modified the text as suggested by the reviewer: “Moreover, specificity of 54 assays was not fully tested in the absence of their respective positive controls. Nevertheless, these designs did not show any cross-reaction with RNA positive controls from other viruses.”
Point #5: In the method section, the authors should state how the batches of infected mosquitoes were prepared.
We added the following information: “Briefly, batches of 60 7-10 day-old females were challenged with an infectious blood meal containing 1.4 mL of washed rabbit erythrocytes, 700 μL of viral suspension and 1 mM of adenosine 5’-triphosphate (ATP) as a phagostimulant (12). The blood meal was provided to mosquitoes at a titer of 107 focus-forming unit (FFU)/mL using a Hemotek membrane feeding system (Hemotek Ltd, Blackburn, UK). After 20 min, fully engorged females were transferred in cardboard containers and maintained with 10% sucrose until examination.”
Reviewer 2 Report
The study ‘A new high-throughput tool to screen mosquito-borne viruses in Zika virus endemic/epidemic areas’ uses valuable mosquito populations native toZika virus endemic/epidemic areas to validate a new method of high throughput virus detection. The manuscript is well written and well referenced. There are a number of concerns particularly concerning presentation and interpretation of the results that require attention.
Concerns
Methods – line 152 – how were the primers specifically designed? Were previously designed/confirmed primers used from other studies? If so reference, if not why not?
The two panels of Figure 1, A and B, are not referenced in the text.
Figure 1B is illegible, even when using zoom function on an e-copy of the manuscript. Figure 1, like Figure 2, needs to have better resolution.
Figure 2 – As the abbreviations within figures are not defined, it is difficult to interpret the data e.g. Fig 2 “67: D4-5” is this a pool of DENV-4? Explanation of abbreviations should be applied to all figures.
The authors identify crossreactivity between a number of species/serotype specific primers. This, however, is not followed up to either redesign the primers or adapt the assay. The authors continue to use the dysfunctional assay in their validation experiments. Can the authors explain why no change to the cross-reactive primers was made? Particularly as the authors suggest an advantage of this technique is “the adaptability of the system by adding new sets of primers and probes targeting newly emergent viruses”.
Line – 230 – “technic”, should be technique?
Line – 248 – provide more description on how YFV was “identified”.
Discussion – line 298 and line 307– the assay is not “unambiguous” as there is cross reactivity. Use less definitive and contradictory language to describe the assay.
The authors provide little discussion of the potential for their PCR confirmation technique to be hindered by a lack of viral dissemination in the mosquito. This should be discussed more.
What is the impact of this technique? The apparatus seems too expensive for developing countries to employ as a sentinel measure and not robust or extensive enough to detect all emerging arboviruses? More detail should be provided in the discussion.
Large sections of the Discussion read more like an introduction (e.g. Line 308-320 and Line 335-341). More emphasis should be placed on discussing the development of the assay, as I have outlined above.
Line 344 “excluding Cx quinquefasciatus..” this statement is too conclusive given the limited data to interpret. Tone down the conclusive language and discuss the potential for dissemination etc. to disrupt PCR confirmation.
Author Response
"A new high-throughput tool to screen mosquito-borne viruses in Zika virus endemic/epidemic areas " (viruses-603136) by Moutailler et al.
Dear Editor,
On behalf of my co-authors, I would like to thank the reviewers for the critical reading of our manuscript. We have considered their requests and made most changes suggested. Please, find below point-by-point answers to reviewers, indicating which changes have been made, and where they have been inserted in the text (highlighted in yellow in the revised version).
Sincerely yours,
Sara Moutailler
Anna-Bella Failloux
Answers to Reviewer 2
Point #1: Methods – line 152 – how were the primers specifically designed? Were previously designed/confirmed primers used from other studies? If so reference, if not why not?
The primers/probes used in this work are new sets developed for our system. Indeed, all the primers/probe used should have a temperature of annealing of 60°C for primers and 70°C for probes and should not cross react with any of all the other viruses targeted. So none of already developed primers/probes available in the literature could have been used directly. We have added the following sentence: “Indeed, selection was based on specific constraints of temperature of annealing (60°C for primers and 70°C for probes); primers/probe sets published in the literature were included if they fit into these criteria.”
Point #2: The two panels of Figure 1, A and B, are not referenced in the text.
We have modified in the new version.
Point #3: Figure 1B is illegible, even when using zoom function on an e-copy of the manuscript. Figure 1, like Figure 2, needs to have better resolution.
Figure 2 – As the abbreviations within figures are not defined, it is difficult to interpret the data e.g. Fig 2 “67: D4-5” is this a pool of DENV-4? Explanation of abbreviations should be applied to all figures.
We agreed with the reviewer’s comment. See also answers to the reviewer 1. So column and row labels of the figure 1 and 2 have been modified accordingly.
Point #4: The authors identify crossreactivity between a number of species/serotype specific primers. This, however, is not followed up to either redesign the primers or adapt the assay. The authors continue to use the dysfunctional assay in their validation experiments. Can the authors explain why no change to the cross-reactive primers was made? Particularly as the authors suggest an advantage of this technique is “the adaptability of the system by adding new sets of primers and probes targeting newly emergent viruses”.
To validate the different sets of primers/probe, we used viral RNA extracted from reference materials. If a cross reactivity was detected, we designed new sets of primers/probe and tested them. In some cases, the new sets did not give better results but we decided to keep them. Handling so many sets without any interference became very quickly difficult to manage. So the list of primers/probe proposed are the best ones we were able to develop with the constraints we have due to the system used.
Point #5: Line – 230 – “technic”, should be technique?
It has been modified in the manuscript.
Point #6: Line – 248 – provide more description on how YFV was “identified”.
This sentence has been removed. The West-African genotype was circulating in Kedougou where our mosquito collections were coming from (Phan et al. Genomic sequence of yellow fever virus from a Dutch traveller returning from the Gambia-Senegal region, the Netherlands, November 2018. Euro Surveill. 2019;24(4)). We failed in isolating the virus.
We also added in Table 1, column “Type of confirmation performed”: “Literature”.
Point #7: Discussion – line 298 and line 307– the assay is not “unambiguous” as there is cross reactivity. Use less definitive and contradictory language to describe the assay. The authors provide little discussion of the potential for their PCR confirmation technique to be hindered by a lack of viral dissemination in the mosquito. This should be discussed more.
What is the impact of this technique? The apparatus seems too expensive for developing countries to employ as a sentinel measure and not robust or extensive enough to detect all emerging arboviruses? More detail should be provided in the discussion.
Large sections of the Discussion read more like an introduction (e.g. Line 308-320 and Line 335-341). More emphasis should be placed on discussing the development of the assay, as I have outlined above.
We agree with the reviewer’s comment. We have added the following sentences on page 16:
“It is widely admitted that viral dissemination beyond the midgut can be a clue attesting the mosquito susceptibility to a virus. However, viral dissemination in mosquitoes depends on mosquito collection date; it increases over time after the infectious blood meal (30). In our study, it was not possible to have information on the physiological age of mosquitoes and when they were infected.” “It is therefore recommended to identify focal points where this technology could be developed and to improve the conditions for transporting biological samples from the field to allow an optimal viral screening.”
Point #8: Line 344 “excluding Cx quinquefasciatus..” this statement is too conclusive given the limited data to interpret. Tone down the conclusive language and discuss the potential for dissemination etc. to disrupt PCR confirmation.
We have replaced “excluding” by “limiting”.
Round 2
Reviewer 2 Report
The authors have addressed my concerns.